# ON SELF MODULATION FOR GENERATIVE ADVERSARIAL NETWORKS

**Ting Chen**[*]
University of California, Los Angeles
tingchen@cs.ucla.edu

**Mario Lucic, Neil Houlsby, Sylvain Gelly**
Google Brain
{lucic,neilhoulsby,sylvaingelly}@google.com

## ABSTRACT

Training Generative Adversarial Networks (GANs) is notoriously challenging. We propose and study an architectural modification, *self-modulation*, which improves GAN performance across different data sets, architectures, losses, regularizers, and hyperparameter settings. Intuitively, self-modulation allows the intermediate feature maps of a generator to change as a function of the input noise vector. While reminiscent of other conditioning techniques, it requires no labeled data. In a large-scale empirical study we observe a relative decrease of $5\% - 35\%$ in FID. Furthermore, all else being equal, adding this modification to the generator leads to improved performance in $124/144$ $(86\%)$ of the studied settings. Self-modulation is a simple architectural change that requires no additional parameter tuning, which suggests that it can be applied readily to any GAN.[1]

## 1 INTRODUCTION

Generative Adversarial Networks (GANs) are a powerful class of generative models successfully applied to a variety of tasks such as image generation (Zhang et al., 2017; Miyato et al., 2018; Karras et al., 2017), learned compression (Tschannen et al., 2018), super-resolution (Ledig et al., 2017), inpainting (Pathak et al., 2016), and domain transfer (Isola et al., 2016; Zhu et al., 2017).

Training GANs is a notoriously challenging task (Goodfellow et al., 2014; Arjovsky et al., 2017; Lucic et al., 2018) as one is searching in a high-dimensional parameter space for a Nash equilibrium of a non-convex game. As a practical remedy one applies (usually a variant of) stochastic gradient descent, which can be unstable and lack guarantees Salimans et al. (2016). As a result, one of the main research challenges is to stabilize GAN training. Several approaches have been proposed, including varying the underlying divergence between the model and data distributions (Arjovsky et al., 2017; Mao et al., 2016), regularization and normalization schemes (Gulrajani et al., 2017; Miyato et al., 2018), optimization schedules (Karras et al., 2017), and specific neural architectures (Radford et al., 2016; Zhang et al., 2018). A particularly successful approach is based on conditional generation; where the generator (and possibly discriminator) are given side information, for example class labels Mirza & Osindero (2014); Odena et al. (2017); Miyato & Koyama (2018). In fact, state-of-the-art conditional GANs inject side information via conditional batch normalization (CBN) layers (De Vries et al., 2017; Miyato & Koyama, 2018; Zhang et al., 2018). While this approach does help, a major drawback is that it requires external information, such as labels or embeddings, which is not always available.

In this work we show that GANs benefit from *self-modulation* layers in the generator. Our approach is motivated by Feature-wise Linear Modulation in supervised learning (Perez et al., 2018; De Vries et al., 2017), with one key difference: instead of conditioning on external information, we condition on the generator's own input. As self-modulation requires a simple change which is easily applicable to all popular generator architectures, we believe that is a useful addition to the GAN toolbox.

**Summary of contributions.** We provide a simple yet effective technique that can added universally to yield better GANs. We demonstrate empirically that for a wide variety of settings (loss

---

[*]Work done at Google.
[1]Code at https://github.com/google/compare_gan

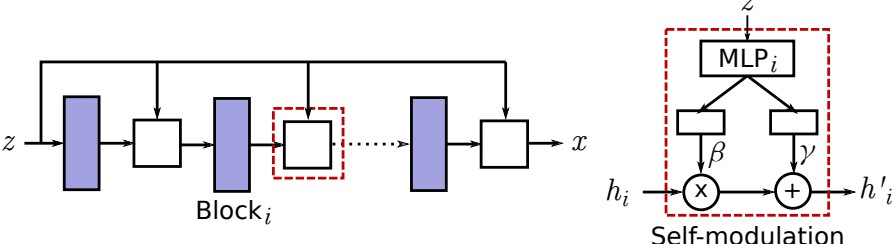

Figure 1: (a) The proposed Self-Modulation framework for a generator network, where middle layers are directly modulated as a function of the generator input $z$. (b) A simple MLP based modulation function that transforms input $z$ to the modulation variables $\beta(z)$ and $\gamma(z)$.

functions, regularizers and normalizers, neural architectures, and optimization settings) that the proposed approach yields between a $5\%$ and $35\%$ improvement in sample quality. When using fixed hyperparameters settings our approach outperforms the baseline in $86\% (124/144)$ of cases. Further, we show that self-modulation still helps even if label information is available. Finally, we discuss the effects of this method in light of recently proposed diagnostic tools, generator conditioning (Odena et al., 2018) and precision/recall for generative models (Sajjadi et al., 2018).

## 2 SELF-MODULATION FOR GENERATIVE ADVERSARIAL NETWORKS

Several recent works observe that conditioning the generative process on side information (such as labels or class embeddings) leads to improved models (Mirza & Osindero, 2014; Odena et al., 2017; Miyato & Koyama, 2018). Two major approaches to conditioning on side information $s$ have emerged: (1) Directly concatenate the side information $s$ with the noise vector $z$ (Mirza & Osindero, 2014), i.e. $z' = [s, z]$. (2) Condition the hidden layers directly on $s$, which is usually instantiated via conditional batch normalization (De Vries et al., 2017; Miyato & Koyama, 2018).

Despite the success of conditional approaches, two concerns arise. The first is practical; side information is often unavailable. The second is conceptual; unsupervised models, such as GANs, seek to model data without labels. Including them side-steps the challenge and value of unsupervised learning.

We propose *self-modulating* layers for the generator network. In these layers the hidden activations are modulated as a function of latent vector $z$. In particular, we apply modulation in a feature-wise fashion which allows the model to re-weight the feature maps as a function of the input. This is also motivated by the FiLM layer for supervised models (Perez et al., 2018; De Vries et al., 2017) in which a similar mechanism is used to condition a supervised network on side information.

Batch normalization (Ioffe & Szegedy, 2015) can improve the training of deep neural nets, and it is widely used in both discriminative and generative modeling (Szegedy et al., 2015; Radford et al., 2016; Miyato et al., 2018). It is thus present in most modern networks, and provides a convenient entry point for self-modulation. Therefore, we present our method in the context of its application via batch normalization. In batch normalization the activations of a layer, $h$, are transformed as

$$h'_\ell = \gamma \odot \frac{h_\ell - \mu}{\sigma} + \beta, \tag{1}$$

where $\mu$ and $\sigma^2$ are the estimated mean and variances of the features across the data, and $\gamma$ and $\beta$ are learnable scale and shift parameters.

**Self-modulation for unconditional (without side information) generation.** In this case the proposed method replaces the non-adaptive parameters $\beta$ and $\gamma$ with input-dependent $\beta(z)$ and $\gamma(z)$, respectively. These are parametrized by a neural network applied to the generator's input (Figure 1). In particular, for layer $\ell$, we compute

$$h'_\ell = \gamma_\ell(z) \odot \frac{h_\ell - \mu}{\sigma} + \beta_\ell(z) \tag{2}$$

Table 1: Techniques for generator conditioning and modulation.

|  | **Only first layer** | **Other Arbitrary layers** |
| --- | --- | --- |
| **Side information $s$** | N/A | Conditional batch normalization (De Vries et al., 2017; Miyato & Koyama, 2018) |
| **Latent vector $z$** | Unconditional Generator (Goodfellow et al., 2014) | (Unconditional) Self-Modulation (this work) |
| **Both $s$ and $z$** | Conditional Generator (Mirza & Osindero, 2014) | (Conditional) Self-Modulation (this work) |

In general, it suffices that $\boldsymbol{\gamma}_\ell(\cdot)$ and $\boldsymbol{\beta}_\ell(\cdot)$ are differentiable. In this work, we use a small one-hidden layer feed-forward network (MLP) with ReLU activation applied to the generator input $z$. Specifically, given parameter matrices $U^{(\ell)}$ and $V^{(\ell)}$, and a bias vector $\boldsymbol{b}^{(\ell)}$, we compute

$$\boldsymbol{\gamma}_\ell(\boldsymbol{z}) = V^{(\ell)} \max(0, U^{(\ell)}\boldsymbol{z} + \boldsymbol{b}^{(\ell)}).$$

We do the same for $\beta(\boldsymbol{z})$ with independent parameters.

**Self-modulation for conditional (with side information) generation.** Having access to side information proved to be useful for conditional generation. The use of labels in the generator (and possibly discriminator) was introduced by Mirza & Osindero (2014) and later adapted by Odena et al. (2017); Miyato & Koyama (2018). In case that side information is available (e.g. class labels $y$), it can be readily incorporated into the proposed method. This can be achieved by simply composing the information $y$ with the input $\boldsymbol{z} \in \mathbb{R}^d$ via some learnable function $g$, i.e. $\boldsymbol{z}' = g(y, \boldsymbol{z})$. In this work we opt for the simplest option and instantiate $g$ as a bi-linear interaction between $\boldsymbol{z}$ and two trainable embedding functions $E, E' : Y \to \mathbb{R}^d$ of the class label $y$, as

$$\boldsymbol{z}' = \boldsymbol{z} + \mathrm{E}(y) + \boldsymbol{z} \odot \mathrm{E}'(y). \tag{3}$$

This conditionally composed $\boldsymbol{z}'$ can be directly used in Equation 1. Despite its simplicity, we demonstrate that it outperforms the standard conditional models.

**Discussion.** Table 1 summarizes recent techniques for generator conditioning. While we choose to implement this approach via batch normalization, it can also operate independently by removing the normalization part in the Equation 1. We made this pragmatic choice due to the fact that such conditioning is common (Radford et al., 2016; Miyato et al., 2018; Miyato & Koyama, 2018).

The second question is whether one benefits from more complex modulation architectures, such as using an attention network (Vaswani et al., 2017) whereby $\beta$ and $\gamma$ could be made dependent on all upstream activations, or constraining the elements in $\boldsymbol{\gamma}$ to $(0, 1)$ which would yield a similar gating mechanism to an LSTM cell (Hochreiter & Schmidhuber, 1997). Based on initial experiments we concluded that this additional complexity does not yield a substantial increase in performance.

## 3 EXPERIMENTS

We perform a large-scale study of self-modulation to demonstrate that this method yields robust improvements in a variety of settings. We consider loss functions, architectures, discriminator regularization/normalization strategies, and a variety of hyperparameter settings collected from recent studies (Radford et al., 2016; Gulrajani et al., 2017; Miyato et al., 2018; Lucic et al., 2018; Kurach et al., 2018). We study both unconditional (without labels) and conditional (with labels) generation. Finally, we analyze the results through the lens of the condition number of the generator's Jacobian as suggested by Odena et al. (2018), and precision and recall as defined in Sajjadi et al. (2018).

### 3.1 EXPERIMENTAL SETTINGS

**Loss functions.** We consider two loss functions. The first one is the non-saturating loss proposed in Goodfellow et al. (2014):

$$V_D(G, D) = \mathbb{E}_{\boldsymbol{x} \sim P_d(\boldsymbol{x})}[\log \sigma(D(\boldsymbol{x}))] + \mathbb{E}_{\boldsymbol{z} \sim P(\boldsymbol{z})}[\log(1 - \sigma(D(G(\boldsymbol{z}))))]$$
$$V_G(G, D) = -\mathbb{E}_{\boldsymbol{z} \sim P(\boldsymbol{z})}[\log \sigma(D(G(\boldsymbol{z})))]$$

The second one is the hinge loss used in Miyato et al. (2018):

$$V_D(G, D) = \mathbb{E}_{\boldsymbol{x} \sim P_d(\boldsymbol{x})}[\min(0, -1 + D(\boldsymbol{x}))] + \mathbb{E}_{\boldsymbol{z} \sim P(\boldsymbol{z})}[\min(0, -1 - D(G(\boldsymbol{z})))]$$
$$V_G(G, D) = -\mathbb{E}_{\boldsymbol{z} \sim P(\boldsymbol{z})}[D(G(\boldsymbol{z}))]$$

**Controlling the Lipschitz constant of the discriminator.** The discriminator's Lipschitz constant is a central quantity analyzed in the GAN literature (Miyato et al., 2018; Zhou et al., 2018). We consider two state-of-the-art techniques: gradient penalty (Gulrajani et al., 2017), and spectral normalization (Miyato et al., 2018). Without normalization and regularization the models can perform poorly on some datasets. For the gradient penalty regularizer we consider regularization strength $\lambda \in \{1, 10\}$.

**Network architecture.** We use two popular architecture types: one based on DCGAN (Radford et al., 2016), and another from Miyato et al. (2018) which incorporates residual connections (He et al., 2016). The details can be found in the appendix.

**Optimization hyper-parameters.** We train all models for 100k generator steps with the Adam optimizer (Kingma & Ba, 2014) (We also perform a subset of the studies with 500K steps and discuss it in. We test two popular settings of the Adam hyperparameters $(\beta_1, \beta_2)$: $(0.5, 0.999)$ and $(0, 0.9)$. Previous studies find that multiple discriminator steps per generator step can help the training (Goodfellow et al., 2014; Salimans et al., 2016), thus we also consider both 1 and 2 discriminator steps per generator step[2]. In total, this amounts to three different sets of hyper-parameters for $(\beta_1, \beta_2, \text{disc\_iter})$: $(0, 0.9, 1)$, $(0, 0.9, 2)$, $(0.5, 0.999, 1)$. We fix the learning rate to 0.0002 as in Miyato et al. (2018). All models are trained with batch size of 64 on a single nVidia P100 GPU. We report the best performing model attained during the training period; although the results follow the same pattern if the final model is report.

**Datasets.** We consider four datasets: CIFAR10, CELEBA-HQ, LSUN-BEDROOM, and IMAGENET. The LSUN-BEDROOM dataset (Yu et al., 2015) contains around 3M images. We partition the images randomly into a test set containing 30588 images and a train set containing the rest. CELEBA-HQ contains 30k images (Karras et al., 2017). We use the $128 \times 128 \times 3$ version obtained by running the code provided by the authors[3]. We use 3000 examples as the test set and the remaining examples as the training set. CIFAR10 contains 70K images ($32 \times 32 \times 3$), partitioned into 60000 training instances and 10000 testing instances. Finally, we evaluate our method on IMAGENET, which contains 1.3M training images and 50K test images. We re-size the images to $128 \times 128 \times 3$ as done in Miyato & Koyama (2018) and Zhang et al. (2018).

**Metrics.** Quantitative evaluation of generative models remains one of the most challenging tasks. This is particularly true in the context of implicit generative models where likelihood cannot be effectively evaluated. Nevertheless, two quantitative measures have recently emerged: The Inception Score and the Frechet Inception Distance. While both of these scores have some drawbacks, they correlate well with scores assigned by human annotators and are somewhat robust.

Inception Score (IS) (Salimans et al., 2016) posits that that the conditional label distribution $p(y|\boldsymbol{x})$ of samples containing meaningful objects should have low entropy, while the marginal label distribution $p(y)$ should have high entropy. Formally, $\text{IS}(G) = \exp(\mathbb{E}_{x \sim G}[\text{d}_{\text{KL}}(p(y|\boldsymbol{x}), p(y)])$. The score is computed using an Inception classifier (Szegedy et al., 2015). Drawbacks of applying IS to model comparison are discussed in Barratt & Sharma (2018).

An alternative score, the Frechet Inception Distance (FID), requires no labeled data (Heusel et al., 2017). The real and generated samples are first embedded into a feature space (using a specific layer of InceptionNet). Then, a multivariate Gaussian is fit each dataset and the distance is computed as $\text{FID}(x, g) = ||\mu_x - \mu_g||_2^2 + \text{Tr}(\Sigma_x + \Sigma_g - 2(\Sigma_x \Sigma_g)^{\frac{1}{2}})$, where $\mu$ and $\Sigma$ denote the empirical mean and covariance and subscripts $x$ and $g$ denote the true and generated data, respectively. FID was shown to be robust to various manipulations and sensitive to mode dropping (Heusel et al., 2017).

---

[2]We also experimented with 5 steps which didn't outperform the 2 step setting.
[3]Available at https://github.com/tkarras/progressive_growing_of_gans.

Table 2: In the unpaired setting (as defined in Section 3.2), we compute the median score (across random seeds) and report the best attainable score across considered optimization hyperparameters. SELF-MOD is the method introduced in Section 2 and BASELINE refers to batch normalization. We observe that the proposed approach outperforms the baseline in 30 out of 32 settings. The relative improvement is detailed in Table 3. The standard error of the median is within 3% in the majority of the settings and is presented in Table 6 for clarity.

| TYPE | ARCH | LOSS | METHOD | BEDROOM | CELEBAHQ | CIFAR10 | IMAGENET |
|---|---|---|---|---|---|---|---|
| GRADIENT PENALTY | RES | HINGE | SELF-MOD | 22.62 | 27.03 | 26.93 | 78.31 |
| | | | BASELINE | 27.75 | 30.02 | 28.14 | 86.23 |
| | | NS | SELF-MOD | 25.30 | 26.65 | 26.74 | 85.67 |
| | | | BASELINE | 36.79 | 33.72 | 28.61 | 98.38 |
| | SNDC | HINGE | SELF-MOD | 110.86 | 55.63 | 33.58 | 90.67 |
| | | | BASELINE | 119.59 | 68.51 | 36.24 | 116.25 |
| | | NS | SELF-MOD | 120.73 | 125.44 | 33.70 | 101.40 |
| | | | BASELINE | 134.13 | 131.89 | 37.12 | 122.74 |
| SPECTRAL NORM | RES | HINGE | SELF-MOD | 14.32 | 24.50 | 18.54 | 68.90 |
| | | | BASELINE | 17.10 | 26.15 | 20.08 | 78.62 |
| | | NS | SELF-MOD | 14.80 | 26.27 | 20.63 | 80.48 |
| | | | BASELINE | 17.50 | 30.22 | 23.81 | 120.82 |
| | SNDC | HINGE | SELF-MOD | 48.07 | 22.51 | 24.66 | 75.87 |
| | | | BASELINE | 38.31 | 27.20 | 26.33 | 90.01 |
| | | NS | SELF-MOD | 46.65 | 24.73 | 26.09 | 76.69 |
| | | | BASELINE | 40.80 | 28.16 | 27.41 | 93.25 |
| BEST OF ABOVE | | | SELF-MOD | **14.32** | **22.51** | **18.54** | **68.90** |
| | | | BASELINE | 17.10 | 26.15 | 20.08 | 78.62 |

## 3.2 ROBUSTNESS EXPERIMENTS FOR UNCONDITIONAL GENERATION

To test robustness, we run a Cartesian product of the parameters in Section 3.1 which results in 36 settings for each dataset (2 losses, 2 architectures, 3 hyperparameter settings for spectral normalization, and 6 for gradient penalty). For each setting we run five random seeds for self-modulation and the baseline (no self-modulation, just batch normalization). We compute the median score across random seeds which results in 1440 trained models.

We distinguish between two sets of experiments. In the *unpaired setting* we define the model as the tuple of loss, regularizer/normalization, neural architecture, and conditioning (self-modulated or classic batch normalization). For each model compute the minimum FID across optimization hyperparameters ($\beta_1$, $\beta_2$, $disc\_iters$). We therefore compare the performance of self-modulation and baseline for each model after hyperparameter optimization. The results of this study are reported in Table 2, and the relative improvements are in Table 3 and Figure 2.

We observe the following: (1) When using the RESNET style architecture, the proposed method outperforms the baseline *in all considered settings*. (2) When using the SNDCGAN architecture, it outperforms the baseline in 87.5% of the cases. The breakdown by datasets is shown in Figure 2. (3) The improvement can be as high as a 33% reduction in FID. (4) We observe similar improvement to the inception score, reported in the appendix.

In the second setting, the *paired setting*, we assess how effective is the technique when simply added to an existing model *with the same set of hyperparameters*. In particular, we fix everything except the type of conditioning – the model tuple now includes the optimization hyperparameters. This results in 36 settings for each data set for a total of 144 comparisons. We observe that self-modulation outperforms the baseline in 124/144 settings. These results suggest that self-modulation can be applied to most GANs even without additional hyperparameter tuning.

**Conditional Generation.** We demonstrate that self-modulation also works for label-conditional generation. Here, one is given access the class label which may be used by the generator and the

Table 3: Reduction in FID over a large class of hyperparameter settings, losses, regularization, and normalization schemes. We observe from 4.3% to 33% decrease in FID. When applied to the RESNET architecture, independently of the loss, regularization, and normalization, SELF-MOD always outperforms the baseline. For SNDCGAN we observe an improvement in 87.5% of the cases (all except two on LSUN-BEDROOM).

| MODEL | | REDUCTION(%) | | MODEL | | REDUCTION(%) | |
|---|---|---|---|---|---|---|---|
| | | RESNET | SNDC | | | RESNET | SNDC |
| HINGE-GP | BEDROOM | 18.50 | 7.30 | NS-GP | BEDROOM | 31.22 | 9.99 |
| | CELEBAHQ | 9.94 | 18.81 | | CELEBAHQ | 20.96 | 4.89 |
| | CIFAR10 | 4.30 | 7.33 | | CIFAR10 | 6.51 | 9.21 |
| | IMAGENET | 9.18 | 22.01 | | IMAGENET | 12.92 | 17.39 |
| HINGE-SN | BEDROOM | 16.25 | -25.48 | NS-SN | BEDROOM | 15.43 | -14.35 |
| | CELEBAHQ | 6.31 | 17.26 | | CELEBAHQ | 13.08 | 12.20 |
| | CIFAR10 | 7.67 | 6.35 | | CIFAR10 | 13.36 | 4.83 |
| | IMAGENET | 12.37 | 15.72 | | IMAGENET | 33.39 | 17.76 |

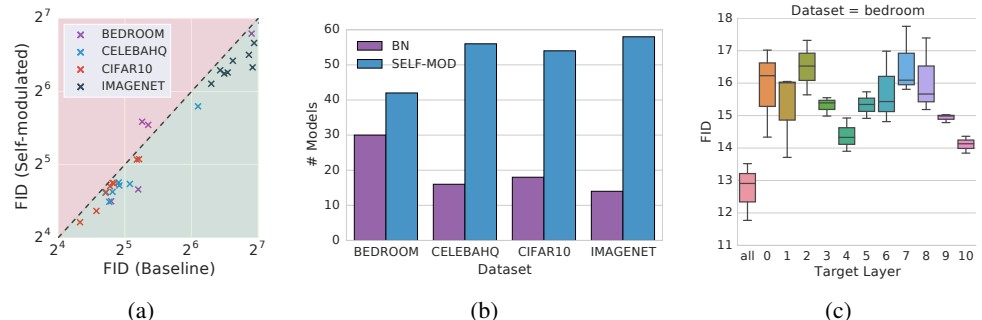

(a)  (b)  (c)

Figure 2: In Figure (a) we observe that the proposed method outperforms the baseline in the unpaired setting. Figure (b) shows the number of models which fall in 80-th percentile in terms of FID (with reverse ordering). We observe that the majority "good" models utilize self-modulation. Figure (c) shows that applying self-conditioning is more beneficial on the later layers, but should be applied to each layer for optimal performance. This effect persists across all considered datasets, see the appendix.

discriminator. We compare two settings: (1) Generator conditioning is applied via label-conditional Batch Norm (De Vries et al., 2017; Miyato & Koyama, 2018) with no use of labels in the discriminator (G-COND). (2) Generator conditioning applied as above, but with projection based conditioning in the discriminator (intuitively it encourages the discriminator to use label discriminative features to distinguish true/fake samples), as in Miyato & Koyama (2018) (P-CGAN). The former can be considered as a special case of the latter where discriminator conditioning is disabled. For P-CGAN, we use the architectures and hyper-parameter settings of Miyato & Koyama (2018). See the appendix, Section B.3 for details. In both cases, we compare standard label-conditional batch normalization to self-modulation with additional labels, as discussed in Section 2, Equation 3.

The results are shown in Table 4. Again, we observe that the simple incorporation of self-modulation leads to a significant improvement in performance in the considered settings.

**Training for longer on IMAGENET.** To demonstrate that self-modulation continues to yield improvement after training for longer, we train IMAGENET for 500k generator steps. Due to the increased computational demand we use a single setting for the unconditional and conditional settings models following Miyato et al. (2018) and Miyato & Koyama (2018), but using only two discriminator steps per generator. We expect that the results would continue to improve if training longer. However, currently results from 500k steps require training for ∼10 days on a P100 GPU.

We compute the median FID across 3 random seeds. After 500k steps the baseline unconditional model attains FID 60.4, self-modulation attains 53.7 (11% improvement). In the conditional setting

Table 4: FID and IS scores in label conditional setting.

| | | UNCONDITIONAL | | G-COND | | P-CGAN | |
| | SCORE | BASELINE | SELF-MOD | BASELINE | SELF-MOD | BASELINE | SELF-MOD |
| --- | --- | --- | --- | --- | --- | --- | --- |
| CIFAR10 | FID | 20.41 | **18.58** | 21.08 | **18.39** | 16.06 | **14.19** |
| IMAGENET | FID | 81.07 | **69.53** | 80.43 | **68.93** | 70.28 | **66.09** |
| CIFAR10 | IS | 7.89 | **8.31** | 8.11 | **8.34** | 8.53 | **8.71** |
| IMAGENET | IS | 11.16 | **12.52** | 11.16 | **12.48** | 13.62 | **14.14** |

self-modulation improves the FID from $50.6$ to $43.9$ ($13\%$ improvement). The improvements in IS are from $14.1$ to $15.1$, and $20.1$ to $22.2$ in unconditional and conditional setting, respectively.

**Where to apply self-modulation?** Given the robust improvements of the proposed method, an immediate question is where to apply the modulation. We tested two settings: (1) applying modulation to every batch normalization layer, and (2) applying it to a single layer. The results of this ablation are in Figure 2. These results suggest that the benefit of self-modulation is greatest in the last layer, as may be intuitive, but applying it to each layer is most effective.

## 4 RELATED WORK

**Conditional GANs.** Conditioning on side information, such as class labels, has been shown to improve the performance of GANs. Initial proposals were based on concatenating this additional feature with the input vector (Mirza & Osindero, 2014; Radford et al., 2016; Odena et al., 2017). Recent approaches, such as the projection cGAN (Miyato & Koyama, 2018) injects label information into the generator architecture using conditional Batch Norm layers (De Vries et al., 2017). Self-modulation is a simple yet effective complementary addition to this line of work which makes a significant difference when no side information is available. In addition, when side information is available it can be readily applied as discussed in Section 2 and leads to further improvements.

**Conditional Modulation.** Conditional modulation, using side information to modulate the computation flow in neural networks, is a rich idea which has been applied in various contexts (beyond GANs). In particular, Dumoulin et al. (2017) apply Conditional Instance Normalization (Ulyanov et al., 2016) to image style-transfer (Dumoulin et al., 2017). Kim et al. (2017) use Dynamic Layer Normalization (Ba et al., 2016) for adaptive acoustic modelling. Feature-wise Linear Modulation (Perez et al., 2018) generalizes this family of methods by conditioning the Batch Norm scaling and bias factors (which correspond to multiplicative and additive interactions) on general external embedding vectors in supervised learning. The proposed method applies to generators in GAN (unsupervised learning), and it works with both unconditional (without side information) and conditional (with side information) settings.

**Multiplicative and Additive Modulation.** Existing conditional modulations mentioned above are usually instantiated via Batch Normalization, which include both multiplicative and additive modulation. These two types of modulation also link to other techniques widely used in neural network literature. The multiplicative modulation is closely related to Gating, which is adopted in LSTM (Hochreiter & Schmidhuber, 1997), gated PixelCNN (van den Oord et al., 2016), Convolutional Sequence-to-sequence networks (Gehring et al., 2017) and Squeeze-and-excitation Networks (Hu et al., 2018). The additive modulation is closely related to Residual Networks (He et al., 2016). The proposed method adopts both types of modulation.

## 5 DISCUSSION

We present a generator modification that improves the performance of most GANs. This technique is simple to implement and can be applied to all popular GANs, therefore we believe that self-modulation is a useful addition to the GAN toolbox.

Our results suggest that self-modulation clearly yields performance gains, however, they do not say how this technique results in better models. Interpretation of deep networks is a complex topic,

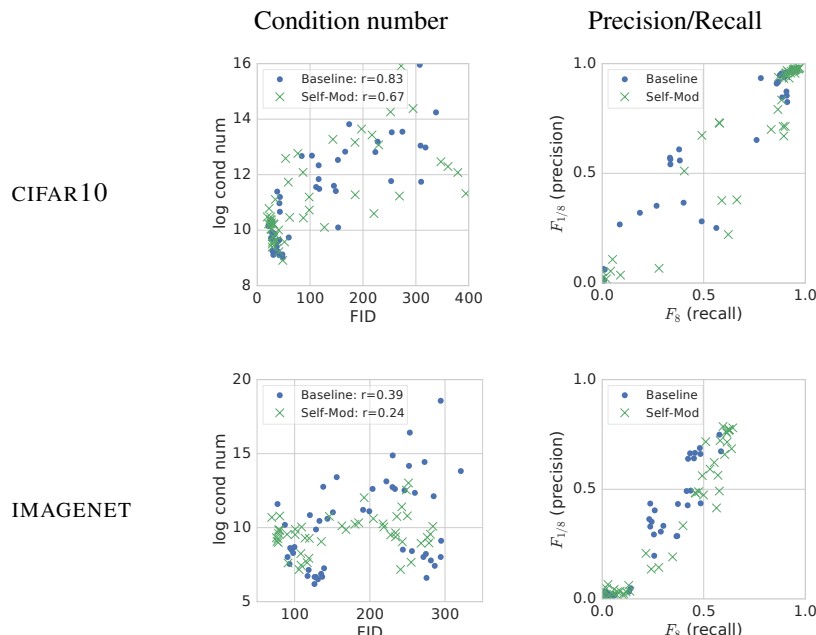

Figure 3: Each point corresponds to a single model/hyperparameter setting. The left-hand plots show the log condition number of the generator versus the FID score. The right-hand plots show the generator precision/recall curves. The $r$ values for the correlation between log condition number and FID on CIFAR10 are $0.67$ and $0.83$ for Self-Mod and Base, respectively. For IMAGENET they are $0.24$ and $0.39$ for Self-Mod and Base, respectively. LSUN-BEDROOM and CELEBA-HQ are in the appendix.

especially for GANs, where the training process is less well understood. Rather than purely speculate, we compute two diagnostic statistics that were proposed recently ignite the discussion of the method's effects.

First, we compute the condition number of the generators Jacobian. Odena et al. (2018) provide evidence that better generators have a Jacobian with lower condition number and hence regularize using this quantity. We estimate the generator condition number in the same was as Odena et al. (2018). We compute the Jacobian $(J_{\boldsymbol{z}})_{i,j} = \frac{\delta G(\boldsymbol{z})_i}{\delta z_j}$ at each $\boldsymbol{z}$ in a minibatch, then average the logarithm of the condition numbers computed from each Jacobian.

Second, we compute a notion of precision and recall for generative models. Sajjadi et al. (2018) define the quantities, $F_8$ and $F_{1/8}$, for generators. These quantities relate intuitively to the traditional precision and recall metrics for classification. Generating points which have low probability under the true data distribution is interpreted as a loss in precision, and is penalized by the $F_8$ score. Failing to generate points that have high probability under the true data distributions is interpreted as a loss in recall, and is penalized by the $F_{1/8}$ score.

Figure 3 shows both statistics. The left hand plot shows the condition number plotted against FID score for each model. We observe that poor models tend to have large condition numbers; the correlation, although noisy, is always positive. This result corroborates the observations in (Odena et al., 2018). However, we notice an inverse trend in the vicinity of the best models. The cluster of the best models with self-modulation has lower FID, but higher condition number, than the best models without self-modulation. Overall the correlation between FID and condition number is smaller for self-modulated models. This is surprising, it appears that rather than unilaterally reducing the condition number, self-modulation provides some training stability, yielding models with a small range of generator condition numbers.

The right-hand plot in Figure 3 shows the $F_8$ and $F_{1/8}$ scores. Models in the upper-left quadrant cover true data modes better (higher precision), and models in the lower-right quadrant produce

more modes (higher recall). Self-modulated models tend to favor higher recall. This effect is most pronounced on IMAGENET.

Overall these diagnostics indicate that self-modulation stabilizes the generator towards favorable conditioning values. It also appears to improve mode coverage. However, these metrics are very new; further development of analysis tools and theoretical study is needed to better disentangle the symptoms and causes of the self-modulation technique, and indeed of others.

ACKNOWLEDGEMENTS

We would like to thank Ilya Tolstikhin for helpful discussions. We would also like to thank Xiaohua Zhai, Marcin Michalski, Karol Kurach and Anton Raichuk for their help with infustrature. We also appreciate general discussions with Olivier Bachem, Alexander Kolesnikov, Thomas Unterthiner, and Josip Djolonga. Finally, we are grateful for the support of other members of the Google Brain team.

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

# A    ADDITIONAL RESULTS

## A.1    INCEPTION SCORES

Table 5: In the unpaired setting (as defined in Section 3.2), we compute the median score (across random seeds) and report the best attainable score across considered optimization hyperparameters. SELF-MOD is the method introduced in Section 2 and BASELINE refers to batch normalization.

| TYPE | ARCH | LOSS | METHOD | BEDROOM | CELEBAHQ | CIFAR10 | IMAGENET |
|------|------|------|--------|---------|----------|---------|----------|
| GRADIENT PENALTY | RESNET | HINGE | SELF-MOD | $5.28 \pm 0.18$ | $2.92 \pm 0.13$ | $7.71 \pm 0.59$ | $11.52 \pm 0.07$ |
| | | | BASELINE | $4.72 \pm 0.11$ | $2.80 \pm 0.08$ | $7.35 \pm 0.02$ | $10.26 \pm 0.09$ |
| | | NS | SELF-MOD | $4.96 \pm 0.17$ | $2.61 \pm 0.05$ | $7.70 \pm 0.05$ | $10.74 \pm 1.20$ |
| | | | BASELINE | $4.54 \pm 0.11$ | $2.60 \pm 0.25$ | $7.26 \pm 0.03$ | $9.49 \pm 0.12$ |
| | SNDCGAN | HINGE | SELF-MOD | $6.34 \pm 0.07$ | $3.05 \pm 0.12$ | $7.37 \pm 0.04$ | $10.99 \pm 0.06$ |
| | | | BASELINE | $5.02 \pm 0.05$ | $3.08 \pm 0.09$ | $6.88 \pm 0.05$ | $8.11 \pm 0.06$ |
| | | NS | SELF-MOD | $6.31 \pm 0.05$ | $3.07 \pm 0.05$ | $7.28 \pm 0.06$ | $10.06 \pm 0.10$ |
| | | | BASELINE | $4.71 \pm 0.05$ | $3.21 \pm 0.20$ | $6.86 \pm 0.06$ | $7.24 \pm 0.16$ |
| SPECTRAL NORM | RESNET | HINGE | SELF-MOD | $3.94 \pm 0.22$ | $3.65 \pm 0.16$ | $8.29 \pm 0.03$ | $12.67 \pm 0.07$ |
| | | | BASELINE | $4.32 \pm 0.17$ | $3.26 \pm 0.16$ | $8.00 \pm 0.03$ | $11.29 \pm 0.12$ |
| | | NS | SELF-MOD | $4.61 \pm 0.18$ | $3.32 \pm 0.09$ | $8.23 \pm 0.04$ | $11.52 \pm 0.28$ |
| | | | BASELINE | $4.07 \pm 0.21$ | $2.58 \pm 0.08$ | $7.93 \pm 0.04$ | $7.40 \pm 0.60$ |
| | SNDCGAN | HINGE | SELF-MOD | $5.85 \pm 0.07$ | $2.74 \pm 0.02$ | $7.90 \pm 0.04$ | $12.50 \pm 0.12$ |
| | | | BASELINE | $4.82 \pm 0.12$ | $2.40 \pm 0.02$ | $7.48 \pm 0.04$ | $9.62 \pm 0.10$ |
| | | NS | SELF-MOD | $5.73 \pm 0.07$ | $2.55 \pm 0.02$ | $7.84 \pm 0.02$ | $11.95 \pm 0.09$ |
| | | | BASELINE | $4.39 \pm 0.14$ | $2.33 \pm 0.01$ | $7.37 \pm 0.04$ | $9.28 \pm 0.13$ |

## A.2    FIDS

Table 6: Table 2 with the standard error of the median.

| TYPE | ARCH | LOSS | METHOD | BEDROOM | CELEBAHQ | CIFAR10 | IMAGENET |
|------|------|------|--------|---------|----------|---------|----------|
| GRADIENT PENALTY | RES | HINGE | SELF-MOD | $22.62 \pm 64.79$ | $27.03 \pm 0.29$ | $26.93 \pm 13.52$ | $78.31 \pm 0.96$ |
| | | | BASE | $27.75 \pm 1.01$ | $30.02 \pm 0.69$ | $28.14 \pm 0.52$ | $86.23 \pm 1.34$ |
| | | NS | SELF-MOD | $25.30 \pm 1.21$ | $26.65 \pm 13.16$ | $26.74 \pm 0.42$ | $85.67 \pm 11.94$ |
| | | | BASE | $36.79 \pm 0.25$ | $33.72 \pm 0.78$ | $28.61 \pm 0.27$ | $98.38 \pm 1.48$ |
| | SNDC | HINGE | SELF-MOD | $110.86 \pm 1.72$ | $55.63 \pm 0.53$ | $33.58 \pm 0.47$ | $90.67 \pm 0.49$ |
| | | | BASE | $119.59 \pm 1.71$ | $68.51 \pm 1.66$ | $36.24 \pm 0.69$ | $116.25 \pm 0.48$ |
| | | NS | SELF-MOD | $120.73 \pm 2.10$ | $125.44 \pm 11.27$ | $33.70 \pm 0.47$ | $101.40 \pm 1.17$ |
| | | | BASE | $134.13 \pm 2.40$ | $131.89 \pm 42.16$ | $37.12 \pm 0.62$ | $122.74 \pm 0.58$ |
| SPECTRAL NORM | RES | HINGE | SELF-MOD | $14.32 \pm 0.40$ | $24.50 \pm 0.46$ | $18.54 \pm 0.15$ | $68.90 \pm 0.67$ |
| | | | BASE | $17.10 \pm 1.44$ | $26.15 \pm 0.70$ | $20.08 \pm 0.31$ | $78.62 \pm 0.97$ |
| | | NS | SELF-MOD | $14.80 \pm 0.40$ | $26.27 \pm 0.48$ | $20.63 \pm 0.20$ | $80.48 \pm 2.43$ |
| | | | BASE | $17.50 \pm 0.64$ | $30.22 \pm 0.48$ | $23.81 \pm 0.17$ | $120.82 \pm 6.82$ |
| | SNDC | HINGE | SELF-MOD | $48.07 \pm 1.77$ | $22.51 \pm 0.38$ | $24.66 \pm 0.40$ | $75.87 \pm 0.37$ |
| | | | BASE | $38.31 \pm 1.42$ | $27.20 \pm 0.80$ | $26.33 \pm 0.54$ | $90.01 \pm 1.06$ |
| | | NS | SELF-MOD | $46.65 \pm 2.72$ | $24.73 \pm 0.25$ | $26.09 \pm 0.19$ | $76.69 \pm 0.89$ |
| | | | BASE | $40.80 \pm 1.75$ | $28.16 \pm 0.17$ | $27.41 \pm 0.43$ | $93.25 \pm 0.35$ |
| BEST OF ABOVE | | | SELF-MOD | **14.32** | **22.51** | **18.54** | **68.90** |
| | | | BASELINE | 17.10 | 26.15 | 20.08 | 78.62 |

## A.3 WHICH LAYER TO MODULATE?

Figure 4 presents the performance when modulating different layers of the generator for each dataset.

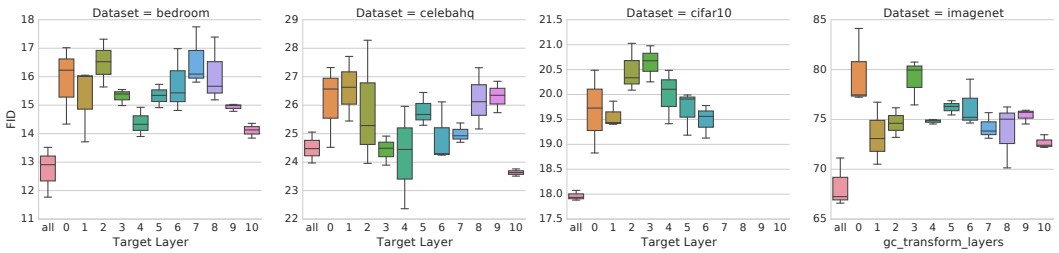

Figure 4: FID distributions resulting from Self-Modulation on different layers.

## A.4 CONDITIONING AND PRECISION/RECALL

Figure 5 presents the generator Jacobian condition number and precision/recall plot for each dataset.

## B MODEL ARCHITECTURES

We describe the model structures that are used in our experiments in this section.

## B.1 SNDCGAN ARCHITECTURES

The SNDCGAN architecture we follows the ones used in Miyato et al. (2018). Since the resolution of images in CIFAR10is $32 \times 32 \times 3$, while resolutions of images in other datasets are $128 \times 128 \times 3$. There are slightly differences in terms of spatial dimensions for both architectures. The proposed self-modulation is applied to replace existing BN layer, we term it sBN (self-modulated BN) for short in Table 7, 8, 9, 10.

## B.2 RESNET ARCHITECTURES

The ResNet architecture we also follows the ones used in Miyato et al. (2018). Again, due to the resolution differences, two ResNet architectures are used in this work. The proposed self-modulation is applied to replace existing BN layer, we term it sBN (self-modulated BN) for short in Table 11, 12, 13, 14.

## B.3 CONDITIONAL GAN ARCHITECTURE

For the conditional setting with label information available, we adopt the Projection Based Conditional GAN (P-cGAN) (Miyato & Koyama, 2018). There are both conditioning in generators as well ad discriminators. For generator, conditional batch norm is applied via conditioning on label information, more specifically, this can be expressed as follows,

$$h'_\ell = \gamma_y \odot \frac{h_\ell - \mu}{\sigma} + \beta_y$$

Where each label $y$ is associated with a scaling and shifting parameters independently.

For discriminator label conditioning, the dot product between final layer feature $\phi(x)$ and label embedding $\mathrm{E}(y)$ is added back to the discriminator output logits, i.e. $D(x, y) = \psi(\phi(x)) + \phi(x)^T \mathrm{E}(y)$ where $\phi(x)$ represents the final feature representation layer of input $x$, and $\psi(\cdot)$ is the linear transformation maps the feature vector into a real number. Intuitively, this type of conditional discriminator encourages discriminator to use label discriminative features to distinguish true/fake samples. Both the above conditioning strategies do not dependent on the specific architectures, and can be applied to above architectures with small modifications.

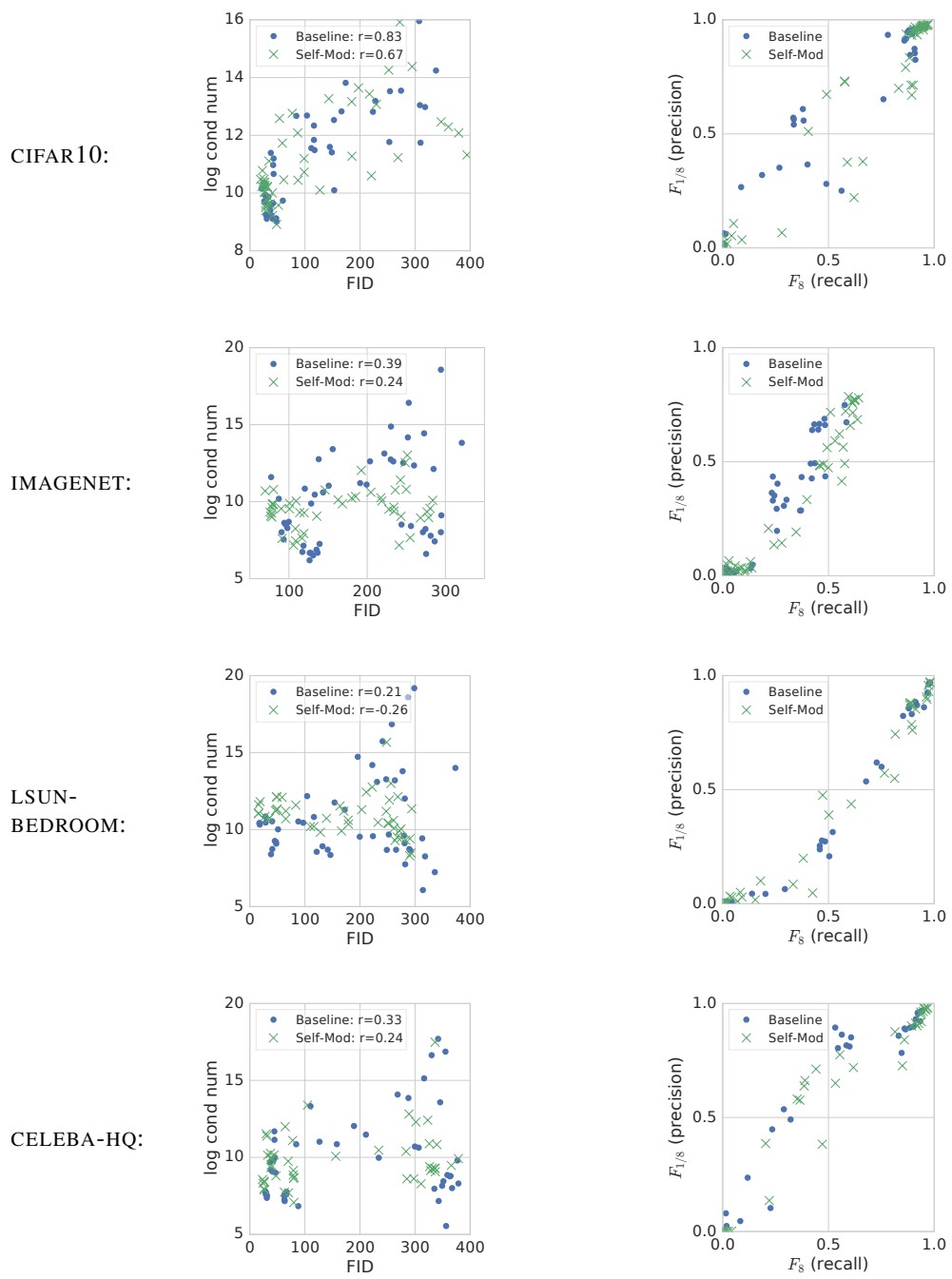

Figure 5: Each point in each plot corresponds to a single model for all parameter configurations. The model with mean FID score across the five random seeds was chosen. The left-hand plots show the log condition number of the generator versus the FID score for each model. The right-hand generator precision/recall metrics.

We use the same architectures and hyper-parameter settings[4] as in Miyato & Koyama (2018). More specifically, the architecture is the same as ResNet above, and we compare in two settings: (1) only

---

[4]With one exception: to make it consistent with previous unconditional settings (and also due to the computation time), instead of running five discriminator steps per generator step, we only use two discriminator steps per generator step.

generator label conditioning is applied, and there is no projection based conditioning in the discriminator, and (2) both generator and discriminator conditioning are applied, which is the standard full P-cGAN.

Table 7: SNDCGAN Generator with $32 \times 32 \times 3$ resolution. sBN denotes BN with self-modulation as proposed.

| Layer | Details | Output size |
|---|---|---|
| Latent noise | $z \sim \mathcal{N}(0, I)$ | 128 |
| Fully Connected | Linear | $2 \cdot 2 \cdot 512$ |
| | Reshape | $2 \times 2 \times 512$ |
| Deconv | sBN, ReLU | $2 \times 2 \times 512$ |
| | Deconv4x4,stride=2 | $4 \times 4 \times 256$ |
| Deconv | sBN, ReLU | $4 \times 4 \times 256$ |
| | Deconv4x4,stride=2 | $8 \times 8 \times 128$ |
| Deconv | sBN, ReLU | $8 \times 8 \times 128$ |
| | Deconv4x4,stride=2 | $16 \times 16 \times 64$ |
| Deconv | sBN, ReLU | $16 \times 16 \times 64$ |
| | Deconv4x4,stride=2 | $32 \times 32 \times 3$ |
| | Tanh | $32 \times 32 \times 3$ |

Table 8: SNDCGAN Discriminator with $32 \times 32 \times 3$ resolution.

| Layer | Details | Output size |
|---|---|---|
| Input image | - | $32 \times 32 \times 3$ |
| Conv | Conv3x3,stride=1 | $32 \times 32 \times 64$ |
| | LeakyReLU | $32 \times 32 \times 64$ |
| Conv | Conv4x4,stride=2 | $16 \times 16 \times 128$ |
| | LeakyReLU | $16 \times 16 \times 128$ |
| Conv | Conv3x3,stride=1 | $16 \times 16 \times 128$ |
| | LeakyReLU | $16 \times 16 \times 128$ |
| Conv | Conv4x4,stride=2 | $8 \times 8 \times 256$ |
| | LeakyReLU | $8 \times 8 \times 256$ |
| Conv | Conv3x3,stride=1 | $8 \times 8 \times 256$ |
| | LeakyReLU | $8 \times 8 \times 256$ |
| Conv | Conv4x4,stride=2 | $4 \times 4 \times 512$ |
| | LeakyReLU | $4 \times 4 \times 512$ |
| Conv | Conv3x3,stride=1 | $4 \times 4 \times 512$ |
| | LeakyReLU | $4 \times 4 \times 512$ |
| Fully connected | Reshape | $4 \cdot 4 \cdot 512$ |
| | Linear | 1 |

Table 9: SNDCGAN Gnerator with $128 \times 128 \times 3$ resolution. sBN denotes BN with self-modulation as proposed.

| Layer | Details | Output size |
|---|---|---|
| Latent noise | $z \sim \mathcal{N}(0, I)$ | 128 |
| Fully Connected | Linear | $8 \cdot 8 \cdot 512$ |
| | Reshape | $8 \times 8 \times 512$ |
| Deconv | sBN, ReLU | $8 \times 8 \times 512$ |
| | Deconv4x4,stride=2 | $16 \times 16 \times 256$ |
| Deconv | sBN, ReLU | $16 \times 16 \times 256$ |
| | Deconv4x4,stride=2 | $32 \times 32 \times 128$ |
| Deconv | sBN, ReLU | $32 \times 32 \times 128$ |
| | Deconv4x4,stride=2 | $64 \times 64 \times 64$ |
| Deconv | sBN, ReLU | $64 \times 64 \times 64$ |
| | Deconv4x4,stride=2 | $128 \times 128 \times 3$ |
| | Tanh | $128 \times 128 \times 3$ |

Table 10: SNDCGAN Discriminator with $128 \times 128 \times 3$ resolution.

| Layer | Details | Output size |
|---|---|---|
| Input image | - | $128 \times 128 \times 3$ |
| Conv | Conv3x3,stride=1 | $128 \times 128 \times 64$ |
| | LeakyReLU | $128 \times 128 \times 64$ |
| Conv | Conv4x4,stride=2 | $64 \times 64 \times 128$ |
| | LeakyReLU | $64 \times 64 \times 128$ |
| Conv | Conv3x3,stride=1 | $64 \times 64 \times 128$ |
| | LeakyReLU | $64 \times 64 \times 128$ |
| Conv | Conv4x4,stride=2 | $32 \times 32 \times 256$ |
| | LeakyReLU | $32 \times 32 \times 256$ |
| Conv | Conv3x3,stride=1 | $32 \times 32 \times 256$ |
| | LeakyReLU | $32 \times 32 \times 256$ |
| Conv | Conv4x4,stride=2 | $16 \times 16 \times 512$ |
| | LeakyReLU | $16 \times 16 \times 512$ |
| Conv | Conv3x3,stride=1 | $16 \times 16 \times 512$ |
| | LeakyReLU | $16 \times 16 \times 512$ |
| Fully connected | Reshape | $16 \cdot 16 \cdot 512$ |
| | Linear | 1 |

Table 11: ResNet Generator with $32 \times 32 \times 3$ resolution. Each ResNet block has a skip-connection that uses upsampling of its input and a 1x1 convolution. sBN denotes BN with self-modulation as proposed.

| Layer | Details | Output size |
|---|---|---|
| Latent noise | $z \sim \mathcal{N}(0, I)$ | 128 |
| Fully connected | Linear | $4 \cdot 4 \cdot 256$ |
| | Reshape | $4 \times 4 \times 256$ |
| ResNet block | sBN, ReLU | $4 \times 4 \times 256$ |
| | Upsample | $8 \times 8 \times 256$ |
| | Conv3x3, sBN, ReLU | $8 \times 8 \times 256$ |
| | Conv3x3 | $8 \times 8 \times 256$ |
| ResNet block | sBN, ReLU | $8 \times 8 \times 256$ |
| | Upsample | $16 \times 16 \times 256$ |
| | Conv3x3, sBN, ReLU | $16 \times 16 \times 256$ |
| | Conv3x3 | $16 \times 16 \times 256$ |
| ResNet block | sBN, ReLU | $16 \times 16 \times 256$ |
| | Upsample | $32 \times 32 \times 256$ |
| | Conv3x3, sBN, ReLU | $32 \times 32 \times 256$ |
| | Conv3x3 | $32 \times 32 \times 256$ |
| Conv | sBN, ReLU | $128 \times 128 \times 3$ |
| | Conv3x3, Tanh | $128 \times 128 \times 3$ |

Table 12: ResNet Discriminator with $32 \times 32 \times 3$ resolution. Each ResNet block has a skip-connection that applies a 1x1 convolution with possible downsampling according to spatial dimension.

| Layer | Details | Output size |
|---|---|---|
| Input image | | $32 \times 32 \times 3$ |
| ResNet block | Conv3x3 | $32 \times 32 \times 128$ |
| | ReLU,Conv3x3 | $32 \times 32 \times 128$ |
| | Downsample | $16 \times 16 \times 128$ |
| ResNet block | ReLU,Conv3x3 | $16 \times 16 \times 128$ |
| | ReLU,Conv3x3 | $16 \times 16 \times 128$ |
| | Downsample | $8 \times 8 \times 128$ |
| ResNet block | ReLU,Conv3x3 | $8 \times 8 \times 128$ |
| | ReLU,Conv3x3 | $8 \times 8 \times 128$ |
| ResNet block | ReLU,Conv3x3 | $8 \times 8 \times 128$ |
| | ReLU,Conv3x3 | $8 \times 8 \times 128$ |
| Fully connected | ReLU,GlobalSum pooling | 128 |
| | Linear | 1 |

Table 13: ResNet Generator with $128 \times 128 \times 3$ resolution. Each ResNet block has a skip-connection that uses upsampling of its input and a 1x1 convolution. sBN denotes BN with self-modulation as proposed.

| Layer | Details | Output size |
|---|---|---|
| Latent noise | $z \sim \mathcal{N}(0, I)$ | 128 |
| Fully connected | Linear | $4 \cdot 4 \cdot 1024$ |
| | Reshape | $4 \times 4 \times 1024$ |
| ResNet block | sBN, ReLU | $4 \times 4 \times 1024$ |
| | Upsample | $8 \times 8 \times 1024$ |
| | Conv3x3, sBN, ReLU | $8 \times 8 \times 1024$ |
| | Conv3x3 | $8 \times 8 \times 1024$ |
| ResNet block | sBN, ReLU | $8 \times 8 \times 1024$ |
| | Upsample | $16 \times 16 \times 1024$ |
| | Conv3x3, sBN, ReLU | $16 \times 16 \times 1024$ |
| | Conv3x3 | $16 \times 16 \times 512$ |
| ResNet block | sBN, ReLU | $16 \times 16 \times 512$ |
| | Upsample | $32 \times 32 \times 512$ |
| | Conv3x3, sBN, ReLU | $32 \times 32 \times 512$ |
| | Conv3x3 | $32 \times 32 \times 256$ |
| ResNet block | sBN, ReLU | $32 \times 32 \times 256$ |
| | Upsample | $64 \times 64 \times 256$ |
| | Conv3x3, sBN, ReLU | $64 \times 64 \times 256$ |
| | Conv3x3 | $64 \times 64 \times 128$ |
| ResNet block | sBN, ReLU | $64 \times 64 \times 128$ |
| | Upsample | $128 \times 128 \times 128$ |
| | Conv3x3, sBN, ReLU | $128 \times 128 \times 128$ |
| | Conv3x3 | $128 \times 128 \times 64$ |
| Conv | sBN, ReLU | $128 \times 128 \times 3$ |
| | Conv3x3, Tanh | $128 \times 128 \times 3$ |

Table 14: ResNet Discriminator with $128 \times 128 \times 3$ resolution. Each ResNet block has a skip-connection that applies a 1x1 convolution with possible downsampling according to spatial dimension.

| Layer | Details | Output size |
|---|---|---|
| Input image | | $128 \times 128 \times 3$ |
| ResNet block | Conv3x3 | $128 \times 128 \times 64$ |
| | ReLU,Conv3x3 | $128 \times 128 \times 64$ |
| | Downsample | $64 \times 64 \times 64$ |
| ResNet block | ReLU,Conv3x3 | $64 \times 64 \times 64$ |
| | ReLU,Conv3x3 | $64 \times 64 \times 128$ |
| | Downsample | $32 \times 32 \times 128$ |
| ResNet block | ReLU,Conv3x3 | $32 \times 32 \times 128$ |
| | ReLU,Conv3x3 | $32 \times 32 \times 256$ |
| | Downsample | $16 \times 16 \times 256$ |
| ResNet block | ReLU,Conv3x3 | $16 \times 16 \times 256$ |
| | ReLU,Conv3x3 | $16 \times 16 \times 512$ |
| | Downsample | $8 \times 8 \times 512$ |
| ResNet block | ReLU,Conv3x3 | $8 \times 8 \times 512$ |
| | ReLU,Conv3x3 | $8 \times 8 \times 1024$ |
| | Downsample | $4 \times 4 \times 1024$ |
| ResNet block | ReLU,Conv3x3 | $4 \times 4 \times 1024$ |
| | ReLU,Conv3x3 | $4 \times 4 \times 1024$ |
| Fully connected | ReLU,GlobalSum pooling | 1024 |
| | Linear | 1 |

