# OpenReview forum: "On Self Modulation for Generative Adversarial Networks"
_ICLR.cc/2019/Conference_

### Official Review · AnonReviewer1 · 2018-11-01

**Rating:** 7
**Confidence:** 4

**Review:**

The paper examines an architectural feature in GAN generators -- self-modulation -- and presents empirical evidence supporting the claim that it helps improve modeling performance. The self-modulation mechanism itself is implemented via FiLM layers applied to all convolutional blocks in the generator and whose scaling and shifting parameters are predicted as a function of the noise vector z. Performance is measured in terms of Fréchet Inception Distance (FID) for models trained with and without self-modulation on a fairly comprehensive range of model architectures (DCGAN-based, ResNet-based), discriminator regularization techniques (gradient penalty, spectral normalization), and datasets (CIFAR10, CelebA-HQ, LSUN-Bedroom, ImageNet). The takeaway is that self-modulation is an architectural feature that helps improve modeling performance by a significant margin in most settings. An ablation study is also performed on the location where self-modulation is applied, showing that it is beneficial across all locations but has more impact towards the later layers of the generator.

I am overall positive about the paper: the proposed idea is simple, but is well-explained and backed by rigorous evaluation. Here are the questions I would like the authors to discuss further:

- The proposed approach is a fairly specific form of self-modulation. In general, I think of self-modulation as a way for the network to interact with itself, which can be a local interaction, like for squeeze-and-excitation blocks. In the case of this paper, the self-interaction allows the noise vector z to interact with various intermediate features across the generation process, which for me appears to be different than allowing intermediate features to interact with themselves. This form of noise injection at various levels of the generator is also close in spirit to what BigGAN employs, except that in the case of BigGAN different parts of the noise vector are used to influence different parts of the generator. Can you clarify how you view the relationship between the approaches mentioned above?
- It’s interesting to me that the ResNet architecture performs better with self-modulation in all settings, considering that one possible explanation for why self-modulation is helpful is that it allows the “information” contained in the noise vector to better propagate to and influence different parts of the generator. ResNets also have this ability to “propagate” the noise signal more easily, but it appears that having a self-modulation mechanism on top of that is still beneficial. I’m curious to hear the authors’ thoughts in this.
- Reading Figure 2b, one could be tempted to draw a correlation between the complexity of the dataset and the gains achieved by self-modulation over the baseline (e.g., Bedroom shows less difference between the two approaches than ImageNet). Do the authors agree with that?

---

> ### Author Response · Authors · 2018-11-12
> **Our response**
>
> We would like to thank the reviewer for the time and useful feedback. Our response is given below.
>
> - Relationship to z-conditioning strategy in BigGAN.
>
> Thanks for pointing out the connection to this concurrent submission. We will discuss the connections in the related work section. The main differences are as follows:
> 1. BigGAN performs conditional generation, whilst we primarily focus on unconditional generation. BigGAN splits the latent vector z and concatenates it with the label embedding, whereas we transform z using a small MLP per layer, which is arguably more powerful. In the conditional case, we apply both additive and multiplicative interaction between the label and z, instead of concatenation as in BigGAN.
> 2. Overall BigGAN focusses on scalability to demonstrate that one can train an impressive model for conditional generation. Instead, we focus on a single idea, and show that it can be applied very broadly. We provide a thorough empirical evaluation across critical design decisions in GANs and demonstrate that it is a robust and practically useful contribution.
>
> - Propagation of signal and ResNets.
>
> Indeed, ResNets provide a skip connection which helps signal propagation. Arguably, self-modulation has a similar effect. However, there are critical differences in these mechanisms which may explain the benefits of self-modulation in a resnet architecture:
> 1. Self-modulation applies a channel-wise additive and multiplicative operation to each layer. In contrast, residual connections perform only an element-wise addition in the same spatial locality. As a result, channel-wise modulation allows trainable re-weighting of all feature maps, which is not the case for classic residual connections.
> 2. The ResNet skip-connection is either an identity function or a learnable 1x1 convolution, both of which are linear. In self-modulation, the connection from z to each layer is a learnable non-linear function (MLP).
>
> - Reading Figure 2b, one could be tempted to draw a correlation between the complexity of the dataset and the gains achieved by self-modulation over the baseline (e.g., Bedroom shows less difference between the two approaches than ImageNet). Do the authors agree with that?
>
> Yes, we notice more improvements on the harder, more diverse datasets. These datasets also have more headroom for improvement.

---

### Official Review · AnonReviewer3 · 2018-11-05
**The paper is mainly empirical**

**Rating:** 5
**Confidence:** 5

**Review:**

This paper proposes a Self-Modulation framework for the generator network in GANs, where middle layers are directly modulated as a function of the generator input z.
Specifically, the method is derived via batch normalization (BN), i.e. the learnable scale and shift parameters in BN are assumed to depend on z, through a small one-hidden layer MLP. This idea is something new, although quite straight-forward.
Extensive experiments with varying losses, architectures, hyperparameter settings are conducted to show self-modulation improves baseline GAN performance.

The paper is mainly empirical, although the authors compute two diagnostic statistics to show the effect of the self-modulation method. It is still not clear why self-modulation stabilizes the generator towards small conditioning values.

The paper presents two loss functions at the beginning of section 3.1 - the non-saturating loss and the hinge loss. It should be pointed out that the D in the hinge loss represents a neural network output without range restriction, while the D in the non-saturating loss represents sigmoid output, limiting to take in [0,1]. It seems that the authors are not aware of this difference.

In addition to report the median scores, standard deviations should be reported.

===========  comments after reading response ===========

I do not see in the updated paper that this typo (in differentiating D in hinge loss and non-saturating loss) is corrected.

Though fundamental understanding can happen asynchronously, I reserve my concern that such empirical method is not substantial enough to motivate acceptance in ICLR, especially considering that in (only) 124/144 (86%) of the studied settings, the results are improved. And there is no analysis of the failure settings.

---

> ### Author Response · Authors · 2018-11-12
> **More fundamental understanding can happen asynchronously, while we presented a careful empirical evaluation**
>
> We would like to thank the reviewer for the time and useful feedback. Our response is given below.
>
> - The paper is mainly empirical, although the authors compute two diagnostic statistics to show the effect of the self-modulation method. It is still not clear why self-modulation stabilizes the generator towards small conditioning values.
>
> We consider self-modulation as an architectural change in the line of changes such as residual connections or gating: simple, yet widely applicable and robust. As a first step, we provide a careful empirical evaluation of its benefits. While we have provided some diagnostics statistics, understanding deeply why this method helps will fuel interesting future research. Similar to residual connections, gating, dropout, and many other recent advances, more fundamental understanding will happen asynchronously and should not gate its adoption and usefulness for the community.
>
> - It should be pointed out that the D in the hinge loss represents a neural network output without range restriction, while the D in the non-saturating loss represents sigmoid output, limiting to take in [0,1]. It seems that the authors are not aware of this difference.
>
> We are aware of this key difference and we apply the sigmoid function to scale the output of the discriminator to the [0,1] range for the non-saturating loss. Thanks for carefully reading our manuscript and noticing this typo which we will correct.
>
> - In addition to report the median scores, standard deviations should be reported.
>
> We omitted standard errors simply to reduce clutter. The standard error of the median is within 3% in the majority of the settings and is presented in both Tables 5 and Table 6.

---

> ### Comment · AnonReviewer1 · 2018-11-27
> **In disagreement**
>
> It appears that Reviewer 2 and I disagree with Reviewer 3 in terms of submission rating. I feel strongly about the submission being publication-worthy, and I would like to challenge Reviewer 2’s score.
>
> There is ample room in a research conference for empirical contributions, provided the experimentation is carried out rigorously. To me, the bar for acceptance for this type of paper is 1) whether or not the results can be expected to generalize outside of the reported experimental setting, 2) whether the proposed approach has the potential to have an impact in the research community, and 3) whether the approach and results are communicated clearly to the target audience. In this instance, criteria 1) and 3) are easily met in my opinion: the breadth of model architectures, regularization techniques, and datasets used for evaluation makes me confident that the observed performance improvements are not a happy accident, and the paper writing was straightforward and easy to follow. For criterion 2), I am of the opinion that although the proposed self-modulation mechanism isn’t likely to drastically change the way we train and think of GANs, it is nevertheless a good addition to the set of architectural features that could facilitate GAN training.
>
> I feel that asking for a fundamental explanation of how self-modulation helps improve performance is an unreasonable bar to set for acceptance. Plenty of architectural features like dropout or batch normalization were poorly understood at the time they were first presented, yet in retrospect had a significant impact in the research community. Likewise, asking for the proposed approach to show an improvement for more than “only” 86% of the evaluation settings is unreasonably strict: I don’t find it surprising that there are instances in which self-modulation does not improve performance, and given these odds I would certainly try the approach on a new dataset and architecture combination.

---

### Official Review · AnonReviewer2 · 2018-11-07
**Simple idea, shown to work in a large number of settings**

**Rating:** 7
**Confidence:** 4

**Review:**

Summary:
The manuscript proposes a modification of generators in GANs which improves performance under two popular metrics for multiple architectures, loss, benchmarks, regularizers, and hyperparameter settings. Using the conditional batch normalization mechanism, the input noise vector is allowed to modulate layers of the generator. As this modulation only depends on the noise vector, this technique does not require additional annotations. In addition to the extensive experimentation on different settings showing performance improvements, the authors also present an ablation study, that shows the impact of the method when applied to different layers.

Strengths:
- The idea is simple. The experimentation is extensive and results are convincing in that they show a clear improvement in performance using the method in a large majority of settings.
- I also like the ablation study showing the impact of the method applied at different layers.

Requests for clarification/additional information:
- I might have missed that, but are the authors offering an interpretation of their observation that the performance of the self-modulation model performs worse in the combination of spectral normalization and the SNDC architecture?
- The ablation study shows that the impact is highest when modulation is applied to the last layer (if only one layer is modulated). It seems modulation on layer 4 comes in as a close second. I am curious about why that might be.
- I would like to see some more interpretation on why this method works.
- Did the authors inspect generated samples of the baseline and the proposed method? Is there a notable qualitative difference?

Overall, the idea is simple, the explanation is clear and experimentation is extensive. I would like to see more commentary on why this method might have long-term impact (or not).

---

> ### Author Response · Authors · 2018-11-12
> **Our response**
>
> We would like to thank the reviewer for the time and useful feedback. Our response is given below.
>
> - Interpretation of self-modulation model performs worse in the combination of spectral normalization and the SNDC architecture.
>
> Overall, self-modulation appears to yield the most consistent improvement for the deeper ResNet architecture, than the shallower, more poorly performing, SNDC architecture. Self-modulation doesn’t help in the SNDC/Spectral Norm setting on the Bedroom data, where the SNDC architecture appears to perform very poorly compared to ResNet. For the other three datasets, self-modulation helps in this setting though.
>
> - The ablation study shows that the impact is highest when modulation is applied to the last layer (if only one layer is modulated). It seems modulation on layer 4 comes in as a close second. I am curious about why that might be.
>
> Figure 4 in the Appendix contains the equivalent of Figure 2(c) for all datasets. Considering all datasets: (1) Adding self-modulation to all layers performs best. (2) In terms of median performance, adding it to the layer farthest from the input is the most effective. We believe that the apparent significance of layer 4 in Figure 2(c) is statistical noise.
>
> - I would like to see some more interpretation on why this method works.
>
> We consider self-modulation as an architectural change in the line of changes such as residual connections or gating: simple, yet widely applicable and robust. As a first step, we provide a careful empirical evaluation of its benefits. While we have provided some diagnostics statistics, understanding deeply why this method helps will fuel interesting future research. Similar to residual connections, gating, dropout, and many other recent advances, more fundamental understanding will happen asynchronously and should not gate its adoption and usefulness for the community.
>
> - Did the authors inspect generated samples of the baseline and the proposed method? Is there a notable qualitative difference?
>
> A 10% change in FID is visually noticeable. However, we note that FID rewards both improvements in sample quality (precision) and mode coverage (recall), as discussed in Sec 5 of [1]. While we can easily assess the former by visual inspection, the latter is extremely challenging. Therefore, an improvement in FID may not always be easily visible, but may indicate a better generative model of the data.
>
> [1] https://arxiv.org/abs/1806.00035
>
> - Overall, the idea is simple, the explanation is clear and experimentation is extensive. I would like to see more commentary on why this method might have long-term impact (or not).
>
> We view this contribution as a simple yet generic architecture modification which leads to performance improvements. Similarly to residual connections, we would like to see it used in GAN generator architectures, and more generally in decoder architectures in the long term.

---

### Meta-Review · Area_Chair1 · 2018-12-13

**Confidence:** 3
**Recommendation:** Accept (Poster)

**Metareview:**

This manuscript proposes an architectural improvement for generative adversarial network that allows the intermediate layers of a generator to be modulated by the input noise vector using conditional batch normalization. The reviewers find the paper simple and well-supported by extensive experimental results. There were some concerns about the impact of such an empirical study. However, the strength and simplicity of the technique means that the method could be of practical interest to the ICLR community.